# Fitting Genomic Prediction Models with Different Marker Effects among Prefectures to Carcass Traits in Japanese Black Cattle

**DOI:** 10.3390/genes14010024

**Published:** 2022-12-22

**Authors:** Shinichiro Ogawa, Yukio Taniguchi, Toshio Watanabe, Hiroaki Iwaisaki

**Affiliations:** 1Graduate School of Agriculture, Kyoto University, Kyoto 606-8502, Japan; 2Division of Meat Animal and Poultry Research, Institute of Livestock and Grassland Science, Tsukuba 305-0901, Japan; 3National Livestock Breeding Center, Fukushima 961-8511, Japan; 4Maebashi Institute of Animal Science, Livestock Improvement Association of Japan, Inc., Maebashi 371-0121, Japan; 5Sado Island Center for Ecological Sustainability, Niigata University, Niigata 952-0103, Japan

**Keywords:** Japanese Black cattle, degree of marbling, carcass weight, genomic prediction, variance component estimation

## Abstract

We fitted statistical models, which assumed single-nucleotide polymorphism (SNP) marker effects differing across the fattened steers marketed into different prefectures, to the records for cold carcass weight (CW) and marbling score (MS) of 1036, 733, and 279 Japanese Black fattened steers marketed into Tottori, Hiroshima, and Hyogo prefectures in Japan, respectively. Genotype data on 33,059 SNPs was used. Five models that assume only common SNP effects to all the steers (model 1), common effects plus SNP effects differing between the steers marketed into Hyogo prefecture and others (model 2), only the SNP effects differing between Hyogo steers and others (model 3), common effects plus SNP effects specific to each prefecture (model 4), and only the effects specific to each prefecture (model 5) were exploited. For both traits, slightly lower values of residual variance than that of model 1 were estimated when fitting all other models. Estimated genetic correlation among the prefectures in models 2 and 4 ranged to 0.53 to 0.71, all <0.8. These results might support that the SNP effects differ among the prefectures to some degree, although we discussed the necessity of careful consideration to interpret the current results.

## 1. Introduction

In genomic prediction (GP) of breeding value, genome-wide single nucleotide polymorphisms (SNPs) have been used as markers in linkage disequilibrium (LD) with quantitative trait loci (QTL). The size of a training population can affect the accuracy of GP [1], while enlarging the size is often challenging. Larger training populations which are provided by merging multiple breeds or subpopulations of a single breed could be alternatives (e.g., [2,3,4]), however, the accuracy of GP even got worse in some cases (e.g., [5,6,7]), possibly due to a lower persistence of LD phase among breeds or subpopulations, which might lead to the difference in allele substitution effects of SNP markers (e.g., [8,9,10]). To tackle this, studies have been conducted to perform GP incorporating sequencing data, which could give information on causal variants (e.g., [11,12,13]) and to develop statistical models for GP with training populations provided by merging multiple breeds or subpopulations of a single breed (e.g., [14,15,16]).

Japanese Black cattle are the primary breed of Wagyu which are the native beef cattle in Japan and are now globally well known for meat qualities such as marbling (e.g., [17,18,19]). In this breed, as one representative genetic evaluation scheme via an efficient restricted maximum likelihood (REML)–empirical best linear unbiased prediction (BLUP) computing procedure [20,21,22], breeding value estimation for several carcass traits has been conducting in each prefecture. This used the carcass records, including the degree of marbling, of marketed fattened animals and deep pedigree information. For the use of commercial SNP markers, previous studies have shown that the genetic characteristics of the Japanese Black cattle subpopulations could be inferred from genotype data on genome-wide SNP markers [23,24,25,26,27], and studies about practical use of GP in Japanese Black cattle has been also conducted (e.g., [28,29,30]). For carcass traits, GP via genomic BLUP (GBLUP) using genomic relationship matrix (G matrix) [31] is one operational scheme, in which fattened animals shipped to carcass markets are used as a large-scale training population [32,33]. Recently, Takeda et al. [33] assessed the performance of GP for carcass traits with fattened animals collected from 18 out of 47 prefectures in Japan as a training population. Statistical models assuming that the same allele substitution effects of SNP markers were shared among all fattened animals have been exploited in previous studies on GP for carcass traits in Japanese Black cattle (e.g., [32,33,34]). On the other hand, Zoda et al. [26] recently revealed a difference concerning the degree of persistence of LD phase among commercial SNP markers between fattened steers marketed into Hyogo prefecture and those marketed into other prefectures including Tottori and Hiroshima. This finding brought a hypothesis that the SNP effects as LD markers are not, at least completely, identical among fattened animals marketed into different prefectures.

Several studies have reported the results of analyzing data generated by sequencer in Japanese Black cattle [35,36,37,38], and a study on utilizing sequence data into the flamework of GP is warranted. For developing a better statistical model for GP of carcass traits in Japanese Black cattle using commercial SNP markers, according to Thomasen et al. [9], Zoda et al. [39] assessed the performance of the statistical model including the covariates based on the results of population structure analysis using the STRUCTURE software [40]. On the other hand, de los Campos and Sorensen [41] showed the idea of separating marker effects into a common part across groups (or subpopulations) and group-specific parts. Here, using carcass records of fattened steers marketed into Tottori, Hiroshima, and Hyogo prefectures, we assessed the performance of the models, which assumed SNP effects differing among the prefectures, for GP of cold carcass weight (CW) and marbling score (MS) in Japanese Black cattle.

## 2. Materials and Methods

### 2.1. Theory

Firstly, let assume the following additive bi-allelic QTL effect model for different groups (denoted as groups 1 and 2):$$\left[\begin{array}{c} q_{1}\\ q_{2} \end{array}\right]=\left[\begin{array}{c} Q_{1}\\ Q_{2} \end{array}\right]\alpha +\left[\begin{array}{c} \varepsilon_{1}\\ \varepsilon_{2} \end{array}\right],$$
where **q** is the vector of phenotypic values; **Q** is the matrix of the genotypes of QTL; **α** is the vector of additive QTL allele substitution effects; **ε** is the vector of non-genetic effects. In this study, we treated the fattened steers marketed into each of the three prefectures (Tottori, Hiroshima, and Hyogo) as different groups. When using genome-wide SNPs in Hardy-Weinberg equilibrium (HWE) as LD markers, the following equation could be provided:$$\left[\begin{array}{c} Q_{1}\\ Q_{2} \end{array}\right]\alpha =\left[\begin{array}{c} 1\\ 1 \end{array}\right]2pa+\left[\begin{array}{c} M_{1}-2p\\ M_{2}-2p \end{array}\right]a+\left[\begin{array}{c} \beta_{1}\\ \beta_{2} \end{array}\right],$$
where **M** is the matrix containing the number of a counted SNP allele (0, 1, or 2); **p** is the vector of the frequency of counted SNP alleles; **a** is the vector of allele substitution effects; **β** is the vector of residual parts not captured by the SNP markers used; and **1** is the vector of ones. Then:$$\begin{array}{cc} \left[\begin{array}{c} q_{1}\\ q_{2} \end{array}\right] & =\left[\begin{array}{c} 1\\ 1 \end{array}\right]2pa+\left[\begin{array}{c} M_{1}-2p\\ M_{2}-2p \end{array}\right]a+\left[\begin{array}{c} \beta_{1}\\ \beta_{2} \end{array}\right]+\left[\begin{array}{c} \varepsilon_{1}\\ \varepsilon_{2} \end{array}\right]\\ & =\left[\begin{array}{c} 1\\ 1 \end{array}\right]\mu +\left[\begin{array}{c} M_{1}-2p\\ M_{2}-2p \end{array}\right]a+\left[\begin{array}{c} e_{1}\\ e_{2} \end{array}\right]=\left[\begin{array}{c} 1\\ 1 \end{array}\right]\mu +\left[\begin{array}{c} g_{1}\\ g_{2} \end{array}\right]+\left[\begin{array}{c} e_{1}\\ e_{2} \end{array}\right] \end{array},$$
where *μ* is the scalar of the intercept; **g** is the vector of genomic breeding values (GBVs); and **e** is the vector of residuals. Now, the vectors **a**, **e**_1_, and **e**_2_ was treated as random, and their expectation and (co) variance structures were assumed to be:$$E\left[\begin{array}{c} a\\ e_{1}\\ e_{2} \end{array}\right]=\left[\begin{array}{c} 0\\ 0\\ 0 \end{array}\right]\;\mathrm{and}\;V\left[\begin{array}{c} a\\ e_{1}\\ e_{2} \end{array}\right]=\left[\begin{array}{ccc} I\sigma_{a}^{2} & 0 & 0\\ 0 & I\sigma_{e}^{2} & 0\\ 0 & 0 & I\sigma_{e}^{2} \end{array}\right],$$
where $\sigma_{a}^{2}$ is the scalar of the variance of each SNP effect; $\sigma_{e}^{2}$ is the scalar of residual variance; and **I** is the identity matrix. Then, the (co)variance of the vectors **q**_1_ and **q**_2_ was:$$\begin{array}{cc} V\left[\begin{array}{c} q_{1}\\ q_{2} \end{array}\right] & =\left[\begin{array}{cc} \left(M_{1}-2p\right){\left(M_{1}-2p\right)}^{\prime }/c & \left(M_{1}-2p\right){\left(M_{2}-2p\right)}^{\prime }/c\\ \left(M_{2}-2p\right){\left(M_{1}-2p\right)}^{\prime }/c & \left(M_{2}-2p\right){\left(M_{2}-2p\right)}^{\prime }/c \end{array}\right]c\sigma_{a}^{2}+V\left[\begin{array}{c} e_{1}\\ e_{2} \end{array}\right]\\ & =\left[\begin{array}{cc} G_{11} & G_{12}\\ G_{21} & G_{22} \end{array}\right]\sigma_{g}^{2}+I\sigma_{e}^{2}=G\sigma_{g}^{2}+I\sigma_{e}^{2} \end{array},$$
where *c* equals $\sum_{i=1}^{n}2p_{i}\left(1-p_{i}\right)$; $\sigma_{g}^{2}$ is the scalar of the genomic variance, or additive genetic variance explained by the SNP markers used; and **G** is the G matrix calculated according to method 1 in VanRaden [31].

Next, according to the idea shown in de los Campos and Sorensen [41], we further assumed that the SNP effects were different between groups at least partly due to lower persistence of LD phase. The SNP effects, **a**, were divided into a common part across groups, **u**, and group-specific ones, **d** [41]:$$\left[\begin{array}{c} Q_{1}\\ Q_{2} \end{array}\right]\alpha =\left[\begin{array}{c} 2p\left(u+d_{1}\right)\\ 2p\left(u+d_{2}\right) \end{array}\right]+\left[\begin{array}{c} M_{1}-2p\\ M_{2}-2p \end{array}\right]u+\left[\begin{array}{c} M_{1}-2p\\ 0 \end{array}\right]d_{1}+\left[\begin{array}{c} 0\\ M_{2}-2p \end{array}\right]d_{2}+\left[\begin{array}{c} \beta_{1}\\ \beta_{2} \end{array}\right]$$

The expectation and (co)variance structures of the vectors **u**, **d**_1_, **d**_2_, **e**_1_, and **e**_2_ were assumed to be:$$E\left[\begin{array}{c} u\\ d_{1}\\ d_{2}\\ e_{1}\\ e_{2} \end{array}\right]=\left[\begin{array}{c} 0\\ 0\\ 0\\ 0\\ 0 \end{array}\right]\;\mathrm{and}\;V\left[\begin{array}{c} u\\ d_{1}\\ d_{2}\\ e_{1}\\ e_{2} \end{array}\right]=\left[\begin{array}{ccccc} I\sigma_{u}^{2} & 0 & 0 & 0 & 0\\ 0 & I\sigma_{d}^{2} & 0 & 0 & 0\\ 0 & 0 & I\sigma_{d}^{2} & 0 & 0\\ 0 & 0 & 0 & I\sigma_{e}^{2} & 0\\ 0 & 0 & 0 & 0 & I\sigma_{e}^{2} \end{array}\right],$$
where $\sigma_{u}^{2}$ is the scalar of the variance of each of the common SNP effects; and $\sigma_{d}^{2}$ is the scalar of the variance of each of the group-specific SNP effects. Then, the (co)variance of the vectors **q**_1_ and **q**_2_ was:$$\begin{array}{cc} V\left[\begin{array}{c} q_{1}\\ q_{2} \end{array}\right] & =\;Gc\sigma_{u}^{2}+\left(\left[\begin{array}{cc} G_{11} & 0\\ 0 & 0 \end{array}\right]+\left[\begin{array}{cc} 0 & 0\\ 0 & G_{22} \end{array}\right]\right)c\sigma_{d}^{2}+I\sigma_{e}^{2}\\ & =\left[\begin{array}{cc} G_{11}\left(\sigma_{g_{1}}^{2}+\sigma_{g_{2}}^{2}\right) & G_{12}\sigma_{g_{1}}^{2}\\ G_{21}\sigma_{g_{1}}^{2} & G_{22}\left(\sigma_{g_{1}}^{2}+\sigma_{g_{2}}^{2}\right) \end{array}\right]+I\sigma_{e}^{2} \end{array},$$
where $\sigma_{g1}^{2}$ equals $c\sigma_{u}^{2}$; and $\sigma_{g2}^{2}$ equals $c\sigma_{d}^{2}$. Values of phenotypic variance, heritability, and genetic correlation between groups can be obtained as $\sigma_{g1}^{2}+\sigma_{g2}^{2}+\sigma_{e}^{2}$, $\left(\sigma_{g1}^{2}+\sigma_{g2}^{2}\right)/\left(\sigma_{g1}^{2}+\sigma_{g2}^{2}+\sigma_{e}^{2}\right)$, and $\sigma_{g1}^{2}/\left(\sigma_{g1}^{2}+\sigma_{g2}^{2}\right)$, respectively (e.g., [10,42,43]). When assuming **u** = **0**, $\sigma_{g1}^{2}$ is zero and then no genetic correlation among the groups is assumed.

### 2.2. Data Analysis

Animal care and use were according to the protocol approved by the Shirakawa Institute of Animal Genetics Animal Care and Use Committee, Nishigo, Japan (ACUCH21-1).

We analyzed the carcass records for 2048 fattened steers collected from Tottori, Hiroshima, and Hyogo prefectures through 2003 to 2014. The data were also analyzed in Zoda et al. [39]. Here, the fattened steers marketed within Tottori, Hiroshima, and Hyogo prefectures are denoted as “To”, “Hi”, and “Hy” steers, respectively. The numbers of the To, Hi, and Hy steers were 1036, 733, and 279. Traits analyzed were CW and MS. MS were evaluated as beef marbling standard at the cross-section between the sixth and seventh ribs of the left side of a cold carcass by official graders according to the carcass grading standards [44]. Table 1 shows the means and standard deviations (SDs) for phenotypic records. It should be noted that the information about pedigree and fattening farms was not available in this study.

Genotype information on 33,059 SNPs with position information (UMD 3.1) and minor allele frequencies > 0.01 in HWE (*p* > 0.001) in the 2048 steers were used. Genomic DNA extraction, SNP genotyping, and missing genotype imputation were conducted following Watanabe [32]. Briefly, extracted DNA samples were genotyped using either the Illumina BovineSNP50 or BovineLD BeadChip. Missing genotype filling for BovineSNP50 data and imputation from BovineLD to BovineSNP50 data were carried out using Beagle 3.3.2 software [45]. For the imputation, BovineSNP50 genotype data obtained from 651 fattened animals (617 steers and 34 females) were used as a haplotype reference population.

According to method 1 in VanRaden [31], three G matrices, denoted as **G**_1_, **G**_2_, and **G**_3_, were calculated:$$G_{1}=[\begin{array}{ccc} G_{To-To} & G_{To-Hi} & G_{To-Hy}\\ G_{Hi-To} & G_{Hi-Hi} & G_{Hi-Hy}\\ G_{Hy-To} & G_{Hy-Hi} & G_{Hy-Hy} \end{array}],\;G_{2}=\left[\begin{array}{ccc} G_{To-To} & G_{To-Hi} & 0\\ G_{Hi-To} & G_{Hi-Hi} & 0\\ 0 & 0 & G_{Hy-Hy} \end{array}\right],\;G_{3}=\left[\begin{array}{ccc} G_{To-To} & 0 & 0\\ 0 & G_{Hi-Hi} & 0\\ 0 & 0 & G_{Hy-Hy} \end{array}\right],$$
where **G***_a-b_* is the submatrix for the steers marketed into prefectures a and b. For example, **G***_To–Hy_* is the submatrix with 1036 rows and 279 columns for the To steers and Hy steers. Allele frequency was calculated using all the 2048 steers. The **G**_1_ matrix was used for the SNP effects common among the steers, the **G**_2_ matrix was for the SNP effects differing between Hy steers and others, the **G**_3_ was for the SNP effects differing among To, Hi, and Hy steers. Figure 1 shows the heatmaps of the elements of **G**_1_ and **G**_3_ matrices. Note that values of the elements of submatrix for the Hy steers, namely **G***_Hy–Hy_*, were higher in average, as reported by Zoda et al. [39].

We assessed the performance of the five models below. The first model (denoted as model 1) was:$$y=Xb+g_{1}+e,$$
where **y** is the vector of phenotypic records; **b** is the vector of main effects of prefecture (Tottori, Hiroshima, and Hyogo) and year at slaughter (through 2003 to 2014), and the partial linear and quadratic covariates of age at slaughter; **g**_1_ is the vector of GBVs with the (co)variance structure of **G**_1_$\sigma_{g1}^{2}$; **e** is the vector of residuals and the (co)variance structure is $I\sigma_{e}^{2}$; and **X** is an incidence matrix for **b**. Previous studies on GP of carcass traits in Japanese Black cattle have also used this kind of statistical model (e.g., [32,33,34]). We also exploited the following model (model 2):$$y=Xb+g_{1}+g_{2}+e,$$
where **g**_2_ is the vector of GBVs with the (co)variance structure of **G**_2_$\sigma_{g2}^{2}$. We also used the model which ignored the term **g**_1_ in model 2 (model 3). We changed the (co)variance structure of **g**_2_ from **G**_2_$\sigma_{g2}^{2}$ to **G**_3_$\sigma_{g2}^{2}$ (model 4). Furthermore, the model ignoring the term **g**_1_ in model 4 was used (model 5). Therefore, when using models 2 and 4, the total GBVs were the sum of **g**_1_ and **g**_2_.

All parameters were estimated via the Bayesian framework using the Gibbs sampler in BGLR package [46]. The default settings were used for the prior distributions and the vectors **g**_1_, **g**_2_, and **e** were assumed to follow multivariate normal distributions. A single chain of 110,000 samples was run, and the first 10,000 samples discarded as burn-in. Samples after burn-in were used with a thinning rate of 10. We assessed the Gibbs sampling chains by visual inspection. Parameter estimates and their standard errors (SEs) were obtained by calculating the means and SDs of the 10,000 posterior samples. Values of deviance information criterion (DIC) [47], estimated SNP effects, and predicted GBVs were compared among the models. Values of the SNP effects were estimated according to previous studies (e.g., [48,49,50]). For example, the SNP effects common among the steers in models 1, 2, and 4 were calculated as follows:$$\frac{{\left(M-2p\right)}^{\prime }G_{1}^{-1}{\hat{g}}_{1}}{\sum_{i=1}^{33,059}2p_{i}\left(1-p_{i}\right)}.$$

## 3. Results

### 3.1. Variance Component Estimation

When using model 1, the estimates of heritability was 0.49 for CW and 0.40 for MS (Table 2). Previous studies [28,39,51], using carcass records of steers marketed into two to five prefectures, estimated the heritability to be ranging from 0.52 to 0.61 for CW and from 0.51 to 0.78 for MS. The estimated heritability for MS in this study was slightly lower than those in these previous studies, possibly because, as well as the difference in the number of the records, the samples used in the previous studies included ones selectively collected for genome-wide association study for MS [28,39,51]. On the other hand, using carcass records of fattened animals collected from 18 prefectures, Takeda et al. [33] estimated the heritability of CW to be 0.41 and that of marbling score to be 0.35, which were both lower than our estimates. Possible reason was the difference in the number of prefectures where the carcass records were collected. Furthermore, information on fattening farms was not available in this and previous studies and the effect of fattening farm could not be considered, which might affect the results [52]. Yao et al. [42] reported that the estimated heritabilities of feed efficiency traits by merging data collected at North America, Netherland, and Scotland were lower than those estimated using each country data separately and then discussed that this phenomenon was due to increased residual variance. Most of the previous studies on GP of carcass traits in Japanese Black cattle used approximately 30,000 SNPs and G matrix calculated by method 1 of VanRaden [31]. Ogawa et al. [34] compared the results for variance component estimation and GP of CW and MS, varying the number of SNPs (approximately 6000, 30,000, and 570,000, corresponding to low-, medium-, and high-density commercial SNP chips) and the G matrix calculation (methods 1 and 2 of VanRaden [31]) and found that the differences were small comparing using medium- and high-density SNP markers and when comparing the methods of G matrix calculation.

The DIC value of model 1 was the highest among the five models for CW and higher than models 2, 3, and 4 for MS (Table 2). For both traits, the DIC value of model 2 was lower than that of model 3, and the value of model 4 was lower than that of model 5. These results may indicate that the SNP effects were not identical, but assumption of no genetic correlation among the groups is too extreme. The DIC value of model 4 was lower than that of model 2, although the estimated genetic correlation in model 4 was nearer to 1 than that in model 2. This might be due to the difference in proportion of carcass records collected from each prefecture; model 2 might show better fitting than model 4, when more carcass records of the fattened animals in Hyogo prefecture were available. Estimated genetic correlation ranged from 0.53 to 0.71, or all <0.8 proposed by Robertson [53]. However, genetic correlations estimated by using models 2 and 4 in this study have a constrain that the genomic variance was the same between the groups. Under this condition, it is unknown that the value of 0.8 could perform as a criterion to judge whether assuming the same SNP effects in two groups is better or not. Yao et al. [42] estimated the genetic correlations for feed efficiency traits in dairy cattle among North America, Netherland, and Scotland to be ranging from 0.36 to 0.47, namely all lower than ours.

Estimated phenotypic variance was almost the same for all models; the heritability was estimated to be slightly higher for the model with lower DIC value. In Yao et al. [42], the model assuming different SNP effects across countries gave lower residual variance and higher heritability. In the framework of multi-breed GP for residual feed intake in cattle, Khansefid et al. [10] showed the tendency that using models assuming different SNP effects between breeds gave lower REML log-likelihood value and residual variance and higher heritability. On the other hand, larger SEs of $\sigma_{g1}^{2}$ and $\sigma_{g2}^{2}$ in models 2 and 4 would reflect the difficulty in separating the SNP effects. Moreover, posterior samples for the two variance components showed the negative correlation (Figure 2). These results might be due to increased model complexity by simultaneously considering two terms relating to GBV, namely **g**_1_ and **g**_2_, in a given model, multicollinearity occurred by using the same SNP markers to consider **g**_1_ and **g**_2_, and the degree of similarity among the three G matrix, or **G**_1_, **G**_2_, and **G**_3_ used in this study.

### 3.2. SNP Effects and Genomic Breeding Values

Within trait, Pearson correlation coefficients of the corresponding SNP effects were >0.9 among the models (Table 3). For example, correlation of the SNP effects specific to Hy steers was ≥0.96 for CW and ≥0.99 for MS among the four models other than model 1. The correlation of SNP effects specific to different groups were low to negligible. For example, the correlation of SNP effects specific to the steers marketed into different prefectures in model 4 ranged from −0.09 to −0.04 for CW and −0.08 to −0.01 for MS. However, the correlation of common SNP effects with those specific to each group was positive, possibly reflecting the difficulty in separating the effects, and values of the correlation was higher when the size of group was larger. For instance, the correlations of common effects with the effects specific to, Hi, and Hy steers in model 4 were 0.65, 0.62, and 0.27, respectively, for CW and 0.78, 0.40, and 0.38, respectively, for MS.

For both traits, Pearson and Spearman’s rank correlations of GBVs for the steers marketed into each prefecture predicted using model 1 were lower with those predicted using model 3 than the correlations with those predicted using model 2 and were lower with those predicted using model 5 than the correlations with those predicted using model 4 (Table 4). These results would reflect the difference in model assumption, model 1 assumed the genetic correlation of 1 among the prefectures, while models 2 and 4 assumed the positive genetic correlation but lower than 1 and models 3 and 5 assumed no genetic correlation among the prefectures. On the other hand, as shown in Figure 3 as an example, the differences in the mean of predicted GBVs among the prefectures were observed. Furthermore, the degree of this difference was varied when fitting different models (Table 5). For example, the mean of predicted GBVs for CW of 279 Hy steers was 68.9 kg and 55.3 kg lower than that of 1036 To steers when using model 1 and model 4, respectively, while that for MS of Hy steers was 0.08 point lower but 0.15 point higher than that of To steers when using model 1 and model 4, respectively. Similar results with larger differences were observed when comparing model 1 which was the model considered the common SNP effects only and models 3 and 5 which were the models ignoring the common SNP effects. The differences in the mean of predicted GBVs among the models were reduced by adding the estimated effects of the prefectures to the corresponding means of predicted GBVs. Zoda et al. [26] also reported a difference in SNP allele frequencies between fattened steers marketed into Hyogo prefecture and those marketed into Tottori and Hiroshima prefectures, which likely affected the results.

## 4. Discussion

In Japanese Black cattle population, the performance of a relatively simple model, such as model 1 in this study, has been investigated in GP using carcass records of fattening animals collected from multiple markets in Japan (e.g., [32,33,34]). Recently, Zoda et al. [26] reported a lower degree of persistence of LD phase between the fattened steers marketed into Hyogo prefecture and those marketed into Tottori and Hiroshima prefectures. According to this finding, we hypothesized that more sophisticated modeling of SNP effects might be required. Then, according to the idea of de los Campos and Sorensen [41], we here attempted to assess the performance of models considering SNP effects differing among the steers marketed into different prefectures (Tottori, Hiroshima, and Hyogo). Except for model 5 in MS, the DIC values were lower when using the models assuming different SNP effects among the steers rather than when using model 1 which is without such assumption (Table 2). Furthermore, models assuming different SNP effects among the prefecture gave slightly decreased residual variance, and the genetic correlation among the prefecture estimated using models 2 and 4 were <0.8 proposed by Robertson [53]. These results might support our hypothesis. On the other hand, it appeared rather difficult to divide SNP effects into common and specific parts (Figure 2, Table 3) and confounding occurred between the effects of prefectures and the means of GBVs of the steers marketed into each prefecture (Table 5). Khansefid et al. [10] pointed out non-additive genetic effects as one of the possible reasons why SNP × breed interactions exist, other than the differences in LD patterns between SNP and QTL among breeds. A few studies reported the results of variance component estimation for carcass traits in Japanese Black cattle using models including non-additive genetic effects [54,55]. Another choice might be a multiple-trait model where carcass traits collected at different prefectures were regarded as genetically different traits. Overall, continued efforts to seek a better statistical model for GP in Japanese Black cattle with a larger training population, as well as to prepare for the use of sequence data in GP, is required.

Zoda et al. [39] assessed the performance of the model for GP considering the results of the STRUCTURE analysis using commercial SNP markers, according to the findings in the previous study [26]. Recently, by simulation using real SNP genotype data from Danish Holstein, Swedish Red, and Danish Jersey cattle purebred and their admixed individuals, Karaman et al. [56] compared the performance of the two model; one including breed proportions inferred using SNP genotypes as covariates and the other considering breed-of-origin effects. Kudinov et al. [57] applied the single-step genomic evaluation with the metafounder approach to the Holstein and Russian Black & White admixed population. On the other hand, we assessed the performance of models assuming the SNP effects differing across the steers marketed into each prefecture. The performance of this type of models has been assessed in livestock populations (e.g., [10,42,43]) as well as crops and human (e.g., [58,59,60]), and Steyn et al. [61] have introduced this concept into the framework of the single-step evaluation. We assumed homogeneous additive genetic variance across the prefectures to avoid over-parameterization, however, assuming heterogeneous variance might be more reasonable [62,63]. Additionally, a genetic correlation constant across the genome was assumed in this study, while there are studies introducing heterogeneous (co)variance patterns across the genome in multi-trait model flamework, which gave improved performance of GP (e.g., [64,65,66]). However, introducing these assumptions further complicates the model and would require a significant number of records to obtain accurate results.

This and previous studies (e.g., [32,33,34]) on GP of carcass traits in Japanese Black cattle with carcass records collected from multiple markets exploited models that consider the effects of prefectures. The historically closed breeding system in Japanese Black cattle, with breeding plans varying from prefecture to prefecture, has brought a subpopulation structure [17], and then prefectures may be roughly divided into those as suppliers of seedstocks including ones such as Tottori, Hiroshima, and Hyogo prefectures, and those as their multipliers [67]. In 1991, genetic evaluation of carcass traits based on pedigree information using mixed model methodology was begun [22], which led to intensive use of frozen semen from fewer elite sires beyond prefectural borders, resulting in an increase in the genetic relationship among subpopulations and a sharp decline in effective population size [68]. The genetic composition of the prefectures including Tottori and Hiroshima prefectures has been penetrated by gene flow due to intensive use of fewer common elite sires across prefectures [23,68], whereas there has been continuing closed breeding in Hyogo prefecture [69]. In many cases, fattened animals are shipped to carcass markets which are in the same as or near prefecture from that the animals are raised in. These facts could affect the genetic composition of the fattened animals marketed into a given prefecture. As additional attempt, when using model 1 but ignoring the effect of prefectures, additive genetic variance and residual variance were estimated to be 1241.0 ± 121.6 and 1210.8 ± 73.8, respectively, for CW and 1.53 ± 0.17 and 2.00 ± 0.11, respectively, for MS, both were greater than those estimated considering the effect of prefectures. This could be also the evidence of confounding between the effects of prefectures and mean GBV. On the other hand, it should be noted that the genetic diversity of commercial populations could change in relatively short time frames, since the sires mated for producing progenies fattened may vary year by year [23,26], which might be also crucial for the performance of GP with fattened animals collected from multiple markets as a larger training population.

Zoda et al. [26] also reported the difference in SNP allele frequencies between the fattened steers marketed into Hyogo prefecture and those marketed into Tottori and Hiroshima prefectures, which might also affect the results obtained in this study. We found that the allele frequencies of the three SNPs previously reported as ones associated with QTL candidate regions for CW, namely *CW-1*, *CW-2*, and *CW-3*, by Nishimura et al. [70] were different between Hy steers and others (Appendix A). The three regions were estimated to be responsible for totally one-third of additive genetic variance for CW in Japanese Black cattle population [70]. The low allele frequency of *CW-3* seems to be because this was detected in a specific line of Japanese Black cattle and is closely related to dysplasia [71,72] and therefore considered rather undesirable. On the other hand, MS is likely an especially highly polygenic trait according to the findings from previous studies (e.g., [28,29,51]). Zoda et al. [26] found that a certain number of SNPs were monomorphic in Hy steers, which were distributed across the genome. Zoda et al. [73] extracted genes within the regions gathering homozygous SNPs in Hy steers and performed gene ontology analysis to the extracted genes, detecting terms possibly relating to meat quality, such as lipid metabolism. Ookura et al. [74] reported that the frequency of favorable alleles of *SCD* (Stearoyl-CoA Desaturase) and *FASN* (Fatty Acid Synthase), genes related to fatty acid composition were very high in fattened animals from Hyogo prefecture.

To account for the differences in allele frequencies and SNP effects across groups, one can assume, for example, the following equation:$$\left[\begin{array}{c} Q_{1}\\ Q_{2} \end{array}\right]\alpha =\left[\begin{array}{c} 2p_{1}d_{1}\\ 2p_{2}d_{2} \end{array}\right]+\left[\begin{array}{c} M_{1}-2p_{1}\\ 0 \end{array}\right]d_{1}+\left[\begin{array}{c} 0\\ M_{2}-2p_{2} \end{array}\right]d_{2}+\left[\begin{array}{c} \beta_{1}\\ \beta_{2} \end{array}\right]$$
where **p**_1_ and **p**_2_ were the vectors of the frequencies of counted alleles in groups 1 and 2, respectively. Now further assume the following mean and (co)variance structures:$$E\left[\begin{array}{c} d_{1}\\ d_{2}\\ e_{1}\\ e_{2} \end{array}\right]=\left[\begin{array}{c} 0\\ 0\\ 0\\ 0 \end{array}\right]\;\mathrm{and}\;V\left[\begin{array}{c} d_{1}\\ d_{2}\\ e_{1}\\ e_{2} \end{array}\right]=\left[\begin{array}{cccc} I\frac{\sigma_{g}^{2}}{c_{1}} & I\frac{\sigma_{g}^{2}}{\sqrt{c_{1}c_{2}}} & 0 & 0\\ I\frac{\sigma_{g}^{2}}{\sqrt{c_{1}c_{2}}} & I\frac{\sigma_{g}^{2}}{c_{2}} & 0 & 0\\ 0 & 0 & I\sigma_{e}^{2} & 0\\ 0 & 0 & 0 & I\sigma_{e}^{2} \end{array}\right],$$
where *c*_1_ equals $\sum_{i=1}^{n}2p_{1i}\left(1-p_{1i}\right)$ and *c*_2_ equals $\sum_{i=1}^{n}2p_{2i}\left(1-p_{2i}\right)$. Therefore:$$V\left[\begin{array}{c} q_{1}\\ q_{2} \end{array}\right]=\left[\begin{array}{cc} \left(M_{1}-2p_{1}\right){\left(M_{1}-2p_{1}\right)}^{\prime }/c_{1} & \left(M_{1}-2p_{1}\right){\left(M_{2}-2p_{2}\right)}^{\prime }/\sqrt{c_{1}c_{2}}\\ \left(M_{2}-2p_{2}\right)\left(M_{1}-2p_{1}\right)/\sqrt{c_{1}c_{2}} & \left(M_{2}-2p_{2}\right){\left(M_{2}-2p_{2}\right)}^{\prime }/c_{2} \end{array}\right]\sigma_{g}^{2}+I\sigma_{e}^{2}=G^{*}\sigma_{g}^{2}+I\sigma_{e}^{2},$$
where **G*** is the G matrix often used for multi-breed GP (e.g., [15,75,76]). Under these assumptions, the genetic correlation between groups is fixed to 1 but the genomic variance is different. On the other hand, in this case, it might be better to perform quality control per group. It should be noted that the discussion above is based on the case where allele frequencies in current population, but not those in base population, are used. Carcass records collected from Hyogo prefecture was available in this study and Zoda et al. [39], but was not used in Takeda et al. [33], while Takeda et al. [33] used the records collected from 18 prefectures with varying the number of records collected from each prefecture. Therefore, it should be noted that an appropriate discrimination of groups in SNP effect modelling might be different, depending on the data structure.

Sophisticated studies on long-term implementation of genomic selection in Japanese Black cattle are warranted. As observed especially in Holstein cattle populations [77,78,79,80,81,82,83], introducing genomic selection might bring more rapid accumulation of inbreeding and then a decrease in genetic diversity. This perspective is important for Japanese Black cattle because, following Nomura et al. [68], the genetic diversity has been already low in this breed. For example, selection intensity is already high for sire selection in Japanese Black cattle population, so the impact of introducing genomic selection on genetic diversity might be more prominent for dam selection. Commercial SNP markers could be available to assess the information on genetic diversity of Japanese Black cattle (e.g., [23,24,25,26,27]), while using imputed genotypes might cause bias in evaluating genomic inbreeding coefficients for some individuals [84,85]. Ogawa et al. [86] reported that the imputation accuracy was high in average for Japanese Black cattle population, but the individual-level performance could be varied. We used SNP markers genotyped using the Illumina BovineSNP50 chip, and therefore, ascertainment bias in the chip might affected the results. Additionally, even if the use of the results of GP is limited to preselection for carcass traits, the impact of selection on genetic evaluation might be considerable (e.g., [87,88,89]). Regarding the difference between the results of selection based on pedigree-based evaluation and that based on genomic-based evaluation, using chickens, Heidaritabar et al. [90] reported that selection pressure is much more locally for GBLUP, resulting in larger allele frequency change, than pedigree-based BLUP. With computer simulation, Liu et al. [91] compared the results of continued selection based on phenotypic values of candidates and results from pedigree-based BLUP, GBLUP, and Bayesian LASSO analyses in terms of the rate of genetic improvement, degree of inbreeding, QTL allele frequency, changes in genetic variance, and hitch-hiking effect. Gómez-Romano et al. [92] proposed the idea to apply optimal contribution selection approach to a specific genomic region. There are studies on partitioning GBVs based on prior information, such as SNP marker location (e.g., [51,93,94]). In this study, models 2 and 4 also gave partitioned GBVs as the terms **g**_1_ and **g**_2_, and this partitioning was according to the finding about the persistence of LD patterns by Zoda et al. [26]. Further study would be beneficial to exploit the results of GP to efficiently improve while considering the genetic diversity of Japanese Black cattle. Moreover, assessing predictive ability of the models should be encouraged by using a larger dataset and the results of routine genetic evaluation based on deep pedigree information, as previous studies did [32,34].

## 5. Conclusions

Here, we fitted statistical models assuming SNP effects differing across the prefectures to the record for CW and MS of Japanese Black fatten steers marketed into Tottori, Hiroshima, and Hyogo prefectures. Except for model 5 in MS, lower DIC values were obtained when using the models assuming different SNP effects than when using model 1 which considered only common SNP effects. Models assuming different SNP effects gave slightly decreased residual variance. Estimated genetic correlations among the prefectures in models 2 and 4 were <0.8. These results could support the validity of assuming the SNP effects differing among the prefectures to some degree. However, careful consideration is required to interpret the current results, from the viewpoints of the difficulty in separating the SNP effects and the possible confounding. Further comprehensive studies to seek a better statistical model for GP of carcass traits in Japanese Black cattle with a larger sized training population, as well as to provide an approach to successfully implement the results of GP into the ongoing selection scheme of this breed, should be encouraged.

## Figures and Tables

**Figure 1 genes-14-00024-f001:**
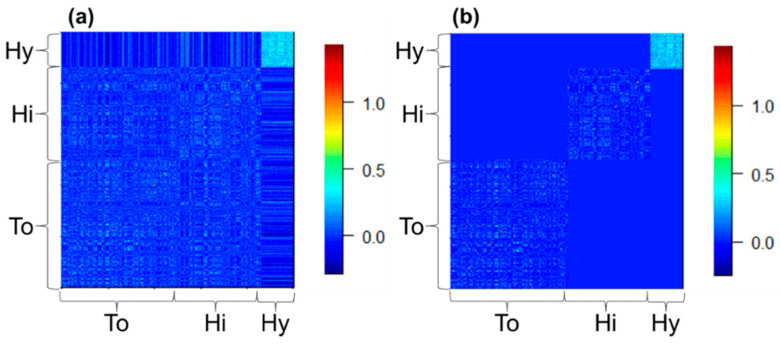
Heatmaps of the two genomic relationship matrices (G matrices). To, Hi, and Hy, the steers marketed within Tottori, Hiroshima, and Hyogo prefectures in Japan. (**a**) The G matrix used to consider allele substitution effects common among To, Hi, and Hy steers (**G**_1_ matrix in the main text); (**b**) The G matrix used to consider the effects differing among To, Hi, and Hy steers (**G**_3_ matrix).

**Figure 2 genes-14-00024-f002:**
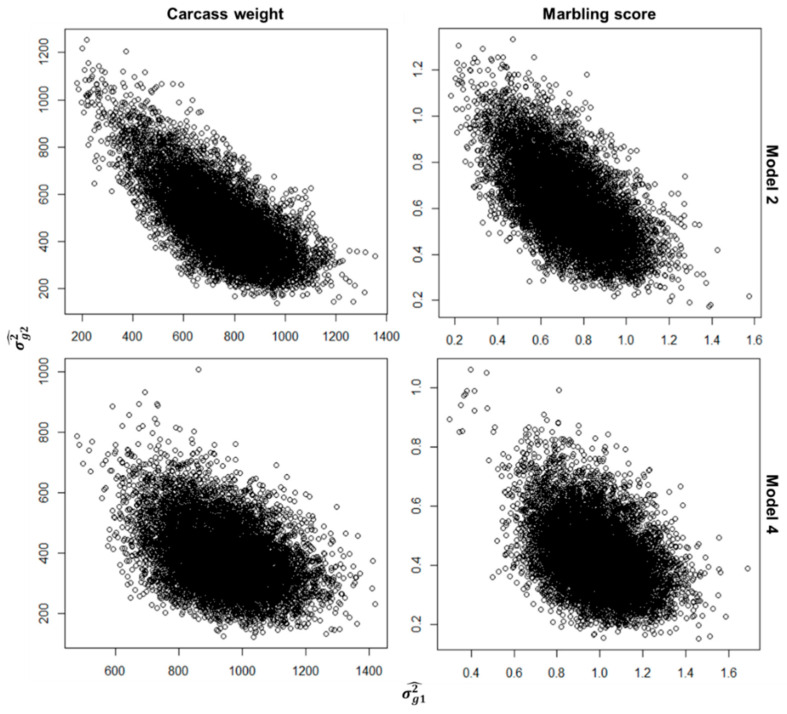
Scatter plots of 10,000 posterior samples for two variance components, $\sigma_{g1}^{2}$ and $\sigma_{g2}^{2}$, in models 2 and 4.

**Figure 3 genes-14-00024-f003:**
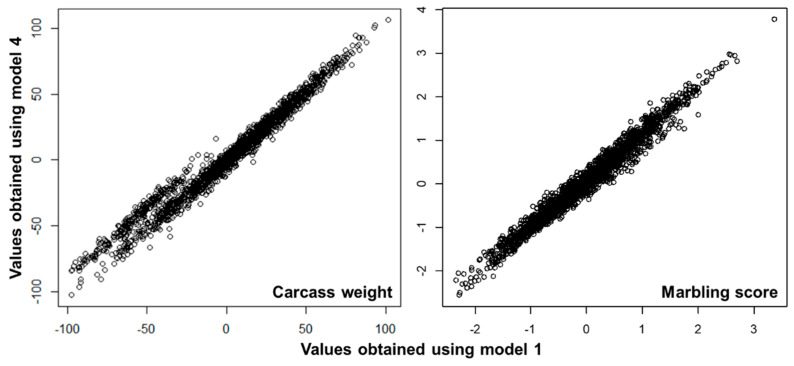
Scatter plots of predicted genomic breeding values obtained using models 1 and 4.

**Table 1 genes-14-00024-t001:** **Number of records (N) and** means and standard deviations (SDs) of phenotypic records.

Item; Unit	Tottori			Hiroshima			Hyogo		
	N	Mean	SD	N	Mean	SD	N	Mean	SD
Age at slaughter; month	1036	28.9	1.2	733	30.0	2.1	279	31.0	1.1
Cold carcass weight; kg		474.2	50.2		485.4	57.1		408.2	39.4
Marbling score; 1 (null) to 12 (very abundance)		5.8	2.0		4.1	1.4		6.5	1.9

**Table 2 genes-14-00024-t002:** Deviance information criterion (DIC) and the estimated parameters and their standard errors (SEs) ^1^.

Model	DIC	$\sigma_{g1}^{2}$		$\sigma_{g2}^{2}$		$\sigma_{e}^{2}$		$\sigma_{p}^{2}$		Heritability		Genetic Correlation	
		Value	SE	Value	SE	Value	SE	Value	SE	Value	SE	Value	SE
		*Cold carcass weight*											
1	0	1163.6	116.2	-	-	1217.7	72.6	2381.3	89.3	0.49	0.04	-	-
2	−25.4	748.2	166.2	485.2	156.4	1173.9	74.8	2407.4	90.8	0.51	0.04	0.61	0.12
3	−14.0	-	-	1211.4	123.6	1183.5	76.9	2394.9	90.0	0.51	0.04	-	-
4	−101.1	941.1	126.9	390.9	105.3	1081.9	79.8	2413.8	90.5	0.55	0.04	0.71	0.07
5	−91.5	-	-	1375.2	137.9	1057.9	88.5	2433.0	90.3	0.56	0.04	-	-
		*Marbling score*											
1	0	1.26	0.15	-	-	1.90	0.10	3.16	0.11	0.40	0.04	-	-
2	−18.8	0.72	0.18	0.63	0.17	1.84	0.10	3.19	0.12	0.42	0.04	0.53	0.12
3	−2.4	-	-	1.28	0.15	1.88	0.11	3.15	0.11	0.40	0.04	-	-
4	−47.0	0.98	0.16	0.43	0.11	1.76	0.11	3.18	0.11	0.44	0.04	0.69	0.08
5	42.4	-	-	1.23	0.15	1.89	0.12	3.13	0.11	0.39	0.04	-	-

^1^ $\sigma_{p}^{2}$, phenotypic variance calculated as $\sigma_{g1}^{2}+\sigma_{g2}^{2}+\sigma_{e}^{2}$. Each DIC value was calculated as the DIC value obtained with the model minus that obtained with model 1.

**Table 3 genes-14-00024-t003:** Pearson correlation coefficients of allele substitution effects of single nucleotide polymorphism (SNP) markers for cold carcass weight and marbling score (above and below diagonals, respectively) ^1^.

Model	Effect	Model 1	Model 2			Model 3		Model 4				Model 5		
		C	C	To-Hi	Hy	To-Hi	Hy	C	To	Hi	Hy	To	Hi	Hy
1	C		1.00	0.95	0.27	0.96	0.31	1.00	0.66	0.61	0.26	0.73	0.67	0.31
2	C	1.00		0.95	0.29	0.96	0.33	1.00	0.65	0.61	0.27	0.73	0.67	0.33
	To-Hi	0.91	0.91		−0.02	1.00	0.02	0.95	0.69	0.64	−0.05	0.75	0.70	0.02
	Hy	0.39	0.40	−0.03		0.02	0.98	0.28	−0.04	−0.02	1.00	0.02	0.02	0.98
3	To-Hi	0.92	0.92	1.00	0.03		0.05	0.96	0.69	0.64	0.00	0.75	0.70	0.07
	Hy	0.41	0.42	0.01	0.99	0.07		0.32	−0.01	0.01	0.96	0.06	0.06	1.00
4	C	0.99	1.00	0.91	0.39	0.92	0.41		0.65	0.62	0.27	0.74	0.68	0.32
	To	0.77	0.77	0.84	0.00	0.84	0.02	0.78		−0.09	−0.01	0.97	0.01	−0.01
	Hi	0.40	0.39	0.46	−0.07	0.45	−0.06	0.40	−0.07		−0.08	0.03	0.97	0.01
	Hy	0.38	0.39	−0.03	1.00	0.01	0.99	0.38	−0.05	−0.04		0.00	0.00	0.96
5	To	0.81	0.81	0.86	0.04	0.87	0.06	0.81	0.99	−0.02	0.03		0.13	0.06
	Hi	0.52	0.52	0.57	0.00	0.57	0.02	0.53	0.10	0.92	−0.01	0.16		0.01
	Hy	0.41	0.42	0.01	0.99	0.05	1.00	0.41	0.02	−0.06	0.99	0.06	0.05	

^1^ C, SNP effects common among all the steers; To, Hi, and Hy, SNP effects specific to the steers marketed into Tottori, Hiroshima, and Hyogo prefectures, respectively; To-Hi, SNP effects specific to the steers marketed into Tottori and Hiroshima prefectures.

**Table 4 genes-14-00024-t004:** Pearson and Spearman’s rank correlations of predicted genomic breeding values obtained using model 1 and those obtained using other 4 models for the steers marketed into Tottori, Hiroshima, and Hyogo prefectures.

Prefecture	Cold Carcass Weight				Marbling Score			
	Model 2	Model 3	Model 4	Model 5	Model 2	Model 3	Model 4	Model 5
	*Pearson correlation*							
Tottori	0.999	0.996	0.992	0.968	0.998	0.993	0.995	0.976
Hiroshima	0.999	0.997	0.991	0.965	0.997	0.990	0.977	0.896
Hyogo	0.982	0.914	0.987	0.912	0.988	0.957	0.994	0.957
	*Spearman’s rank correlation*							
Tottori	0.999	0.996	0.992	0.968	0.998	0.993	0.995	0.976
Hiroshima	0.999	0.997	0.990	0.961	0.997	0.990	0.977	0.892
Hyogo	0.980	0.912	0.985	0.910	0.987	0.957	0.993	0.957

**Table 5 genes-14-00024-t005:** Means and standard deviations (SDs) of predicted genomic breeding values for the steers marketed into each prefecture ^1^.

Model	Tottori		Hiroshima		Hyogo		Mean + Effect of Prefecture		
	Mean	SD	Mean	SD	Mean	SD	Tottori	Hiroshima	Hyogo
	*Cold carcass weight*								
1	0	29.6	−10.1	32.2	−68.9	18.2	0	7.2	−102.7
2	0	30.2	−9.8	32.9	−42.7	18.4	0	7.1	−102.1
3	0	29.9	−9.4	32.5	−5.8	17.3	0	7.7	−102.2
4	0	31.3	−10.8	34.2	−55.3	19.6	0	7.7	−98.8
5	0	31.1	−10.8	33.8	−5.9	19.0	0	9.2	−96.9
	*Marbling score*								
1	0	0.93	−0.36	0.76	−0.08	0.78	0	−1.90	0.59
2	0	0.96	−0.36	0.78	0.33	0.83	0	−1.92	0.61
3	0	0.93	−0.35	0.76	0.60	0.79	0	−1.90	0.59
4	0	1.03	−0.39	0.72	0.15	0.86	0	−1.88	0.72
5	0	0.94	−0.34	0.59	0.54	0.78	0	−1.82	0.71

^1^ Each value was calculated as the value obtained for each prefecture minus that obtained for Tottori prefecture to facilitate comparison of the results.

## Data Availability

The data supporting the findings of this study is shown in the manuscript and Appendix A. Raw phenotype and genotype data sharing is not applicable to this article.

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
