# Peer review of "Fitting Genomic Prediction Models with Different Marker Effects among Prefectures to Carcass Traits in Japanese Black Cattle"

_genes, 2022, doi:10.3390/genes14010024_

Round 1

Reviewer 1 Report

The manuscript reported a genomic prediction study on carcass weight (CW) and marbling score (MS) in three populations of Japanese Black cattle from three prefectures (Tottori, Hiroshima, and Hyogo).  The authors estimated the heritability values and predicted genomic breeding values under five statistical models assuming different SNP effects (Models 2, 3, 4, and 5) or no different SNP effects (Model 1). The obtained results revealed that the models considering different SNP effects fitted better the data. In general, the manuscript is well-written and deserves publication. However, several issues should be addressed in the revision.

Major issues:

1. The authors wrote a general discussion in the section of Results and Discussion, which looked a bit strange. I strongly suggest the authors separate the results and discussion.

2. The authors used imputation to fill in the missing genotypes using BovineSNP50. This can bring secondary ascertainment bias.  And thus the authors should be cautious when interpreting the results.

Minor issues:

3. The authors did not use the same decimal places across the tables.  For instance, all SEs of hereditability in Table 2 was 0.04. I would suggest the authors keep three decimal places in all tables as possible.

4. Why the means of predicted genomic breeding values for Tottori were zero (0) under five models? Did something go wrong?    

Reviewer 2 Report

The present manuscript (Fitting Genomic Prediction Models with Different Marker Effects among prefectures to Carcass Traits in Japanese Black Cattle) tested a novel theory by considering subpopulations in variance component estimation of carcass traits in Japanese Black Cattle. The importance of this study, theory, methods, and results are well presented. However, there are some minor suggestions:

Line 45-49: this is a long sentence. It would be better to summarize it.

Line 63: Please change “prefectures including Tottori and Hiroshima prefectures.” to “prefectures including Tottori and Hiroshima.”

Adding a principle component analysis (PCA) plot to figure 1 for these populations might be helpful to have a perspective on genetic distances.

I would like to suggest testing the predictive ability of models you used. I wonder how the assumptions you used in models can affect prediction accuracy. 
